# History-based action selection bias in posterior parietal cortex

Eun Jung Hwang[1], Jeffrey E. Dahlen[1], Madan Mukundan[1] & Takaki Komiyama [1,2]

Making decisions based on choice-outcome history is a crucial, adaptive ability in life. However, the neural circuit mechanisms underlying history-dependent decision-making are poorly understood. In particular, history-related signals have been found in many brain areas during various decision-making tasks, but the causal involvement of these signals in guiding behavior is unclear. Here we addressed this issue utilizing behavioral modeling, two-photon calcium imaging, and optogenetic inactivation in mice. We report that a subset of neurons in the posterior parietal cortex (PPC) closely reflect the choice-outcome history and history-dependent decision biases, and PPC inactivation diminishes the history dependency of choice. Specifically, many PPC neurons show history- and bias-tuning during the inter-trial intervals (ITI), and history dependency of choice is affected by PPC inactivation during ITI and not during trial. These results indicate that PPC is a critical region mediating the subjective use of history in biasing action selection.

---

[1] Neurobiology Section, Center for Neural Circuits and Behavior, Department of Neurosciences, University of California, San Diego, La Jolla, CA 92093, USA. [2] JST, PRESTO, University of California, San Diego, La Jolla, CA 92093, USA. Eun Jung Hwang and Jeffrey E. Dahlen contributed equally to this work. Correspondence and requests for materials should be addressed to E.J.H. (email: eunjunghwang.phd@gmail.com) or to T.K. (email: tkomiyama@ucsd.edu)

I magine you are deciding on a meal to order at your favorite restaurant. If you enjoyed the dish you ordered the last time you dined there, you may be more inclined to order it again. Such a decision bias shaped by the choice-outcome history can allow one to infer the rules of the environment and generate adaptive behavioral strategies[1–4]. History-dependent biases are not limited to explicitly adaptive contexts such as dynamic foraging tasks. Instead human and animal subjects show diverse idiosyncratic history-dependent biases (e.g., win-stay, lose-switch, or lose-stay) even when the optimal choice is a strict function of environmental stimuli independent of the subject's history, if they are unaware of such a rule or the stimuli are difficult to decipher[5–10]. The prevalent history-dependency of decisions suggests that tracking choice-outcome history to form subjective bias is a fundamental aspect of decision-making, yet the neural circuits mediating this process are largely unknown.

Responses of neurons in the sensorimotor pathway including areas implicated for decision-making show degrees of variability even for identical sensory inputs and motor outputs[11, 12]. Such variability is often treated as random noise[13–15]. However, for future decisions to be biased by history, it is necessary that neural responses are modulated by history, accounting for some of the observed neural variability. Indeed, choice-outcome history has been shown to modulate the activity in a variety of brain areas, including parietal, prefrontal and premotor cortex, and subcortical structures[1, 4, 16–20]. For example, neurons in the lateral intraparietal area of the monkey posterior parietal cortex (PPC) that has been implicated for the accumulation of sensory evidence are modulated by the choice and outcome information of recent trials[19, 21, 22]. However, history affects action selection biases in flexible and complex ways that vary over time and across individuals[1, 19], and it is unclear how the history signals in the brain may account for such a flexible relationship between history and decision bias. Furthermore, these brain areas often contain

intermingled neurons with diverse temporal activity profiles[19, 23, 24], and the roles of specific temporal windows of history-related activity cannot be accessed with traditional lesion or pharmacological inactivation approaches.

Here we combine behavioral modeling, two-photon calcium imaging, and temporally precise inactivation to explore the mechanisms of the subjective, history-dependent decision bias in mice performing a visually instructed action selection task. Similar to previous findings in difficult decision-making tasks, our behavioral model identifies diverse, idiosyncratic relationships between choice-outcome history and action selection bias[5, 7]. These idiosyncratic biases are highly correlated with the pre-stimulus activity of a subset of neurons in PPC. Temporally precise inactivation reveals a causal role of the pre-stimulus activity of PPC, but not the subsequent activity following stimulus onset, in action biases. Therefore, we conclude that PPC is involved in subjective uses of history in biasing action selection.

## Results

**Visually instructed action selection task for head-fixed mice.**
We developed a task in which head-fixed mice moved a joystick with their left forelimb in one of two directions in response to visual cues (Fig. 1a and Supplementary Fig. 1a). In each trial, one of two visual stimuli (gratings moving forward or downward) was presented for one second. This stimulus period was followed by a 2 s memory period. The end of the memory period triggered an auditory go cue, and mice were required to move the joystick in the remembered direction of the visual stimulus to receive a water reward. Trials with no movements, movements before the go cue, and movements in the wrong direction were not rewarded. Mice performed one session per day, $281 \pm 55$ trials per session.

After 2–4 months of incremental training (Supplementary Fig. 1b and Methods section), mice achieved a plateau level of

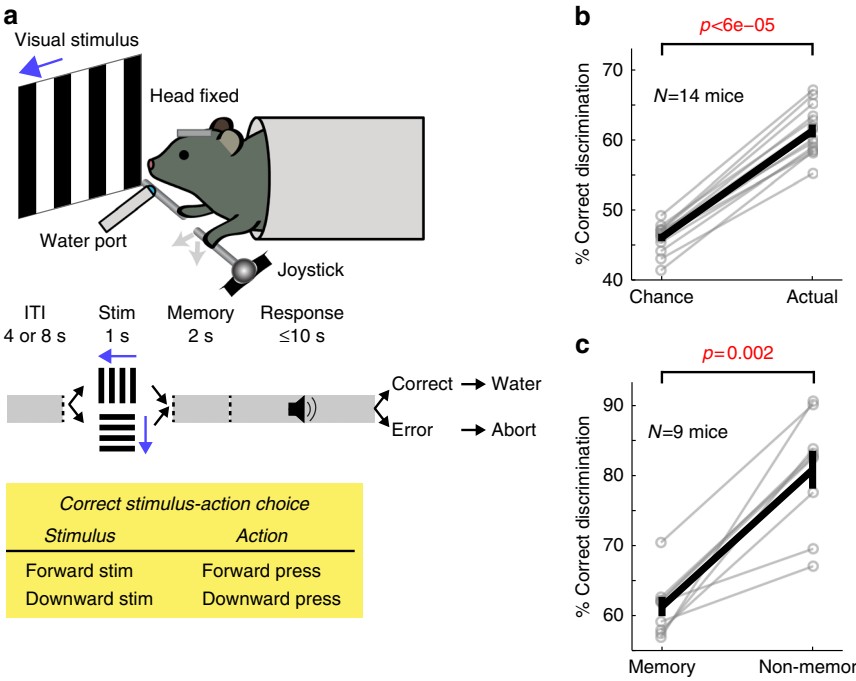

**Fig. 1** Task. **a** Top: task schematic. Middle: task trial structure. Bottom: stimulus-action-outcome rule. **b** Fraction of correctly discriminating trials is significantly greater than chance. Black, mean ± s.e.m. across mice; gray, individual mice. Wilcoxon one-sided signed rank test. **c** Fraction of correctly discriminating trials is significantly greater in non-memory trials (visual stimulus stays on throughout the trial) when randomly interleaved with memory trials. Black, mean ± s.e.m. across mice; gray, individual mice. Wilcoxon one-sided signed rank test

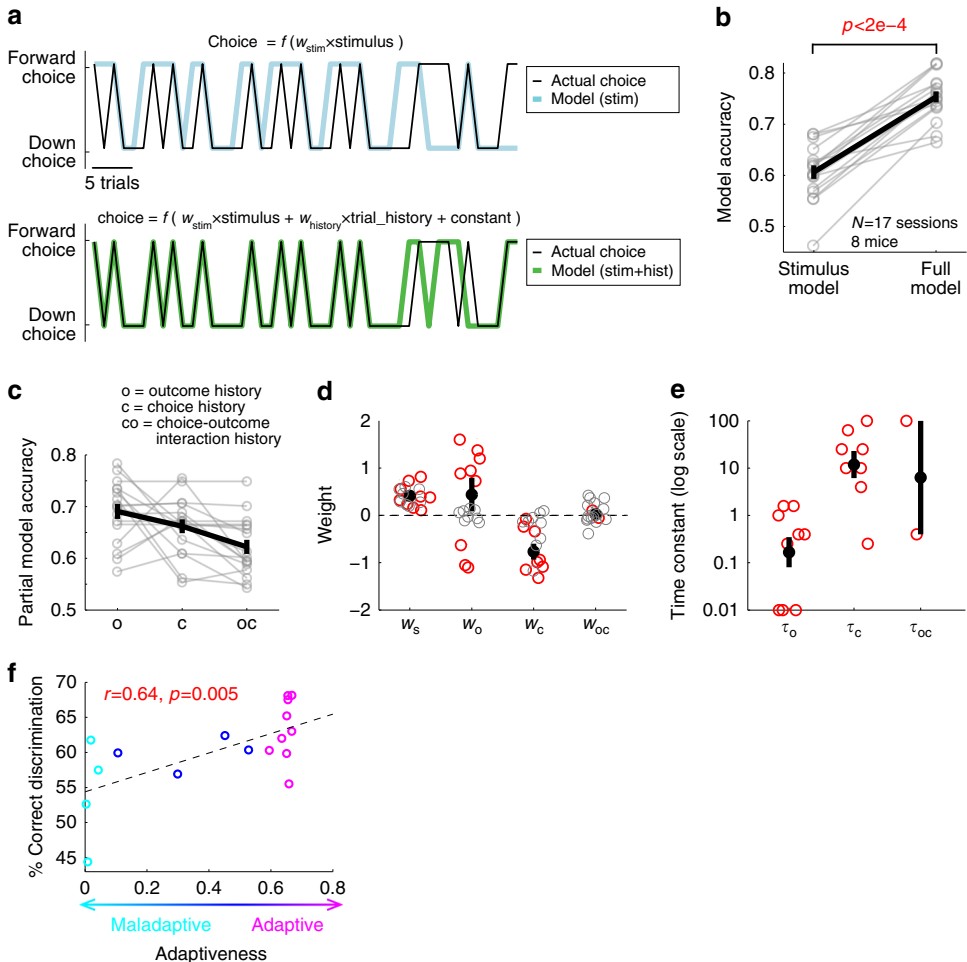

**Fig. 2** Choice-outcome history biases future decision, driving choice variability. **a** Top: example trial-by-trial sequence of an animal's choice (black) with partial model fit using stimulus information only (cyan). Bottom: choice sequence from top plot (black) with full model fit using stimulus, trial history, and a constant (green). **b** Full model including both stimulus and history information predicts choice more accurately than partial model with stimulus only. Black, mean ± s.e.m. across sessions; gray, individual sessions. Wilcoxon one-sided signed rank test. **c** Partial model accuracy using only one history variable (outcome, choice, or outcome-choice interaction) at a time, indicating that outcome history and choice history have large contributions to model accuracy. Black, mean ± s.e.m. across sessions; gray, individual sessions. **d** Weight of each variable. Red circles indicate sessions in which the corresponding partial model (with the variable and a constant) is significantly better than the model with only a constant (likelihood-ratio test between the partial and constant models, $p < 0.05$), while gray circles indicate the other sessions. Black, mean ± s.e.m. across significant sessions. **e** Time constant of each history variable. Sessions are included only if the corresponding partial model (with one history variable and a constant) is significantly better than the model with only a constant (likelihood-ratio test between the partial and constant models, $p < 0.05$). Black, mean ± s.e.m. across sessions; gray, individual sessions. **f** History-dependent strategies are classified as maladaptive (cyan), neutral (blue), or adaptive (magenta) depending on the fraction of correctly discriminating trials achieved by the estimated history models (using only history but not stimulus) relative to chance ($p < 0.05$, Experimental Procedures). Sessions with adaptive strategies tend to be associated with better behavioral performance (Pearson correlation coefficient = 0.64, $p < 0.005$)

performance which was significantly above chance, but far from perfect (Fig. 1b). The behavioral performance was not limited by a difficulty in visual discrimination, as mice performed significantly better in another version of the task without the memory period (Fig. 1c and Supplementary Fig. 2a). The suboptimal performance in the memory task, characterized by variable responses to the same visual stimulus in individual trials, is essential for uncovering internal biases underlying choice variability as shown below.

**History-based bias revealed by behavioral modeling.** We hypothesized that the choice variability in individual trials reflects a systematic fluctuation of hidden internal biases that are shaped by the recent history of the mice, rather than a random

fluctuation due to neural noise, similar to previous findings in difficult decision-making tasks[5, 7, 9]. To test this hypothesis, we built a logistic regression model of the behavior. A similar model has been previously described[7, 25]. Briefly, this model predicts the choice of each mouse on individual trials by utilizing the sensory stimulus of the current trial, the choice-outcome history from previous trials, and a constant choice preference (Eq. 1 and Methods section). Accordingly, the portion of the equation excluding the current stimulus corresponds to the estimate of the internal bias on each trial. Regression was performed in each session independently to identify the weight of each term and the time constants of history terms that best fit the behavior. To avoid overfitting, the accuracy of the model was quantified in a cross validated manner in which the model was built using a fraction of the trials in the session ('training set') and evaluated for the

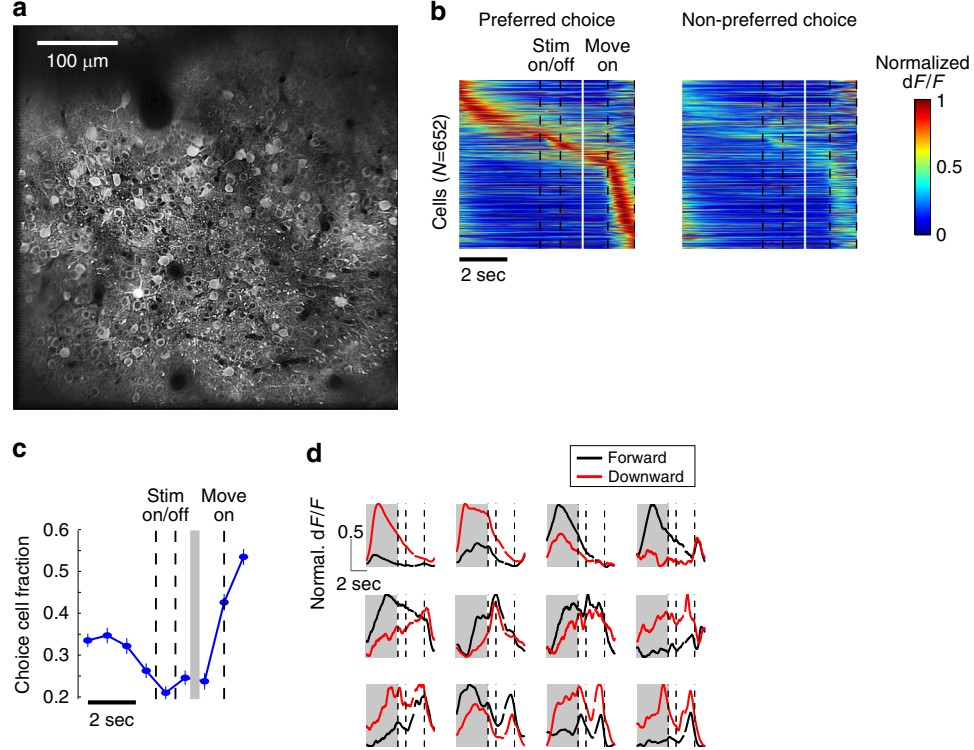

**Fig. 3** Pre-stimulus activity during the inter-trial interval (ITI) in posterior parietal cortex (PPC) predicts the upcoming choice. **a** Two-photon calcium image of PPC neurons. **b** Trial-average activity of PPC choice-selective neurons for preferred vs. non-preferred choice trials. Each row represents the normalized activity of a single neuron sorted by its peak activity timing. **c** Fraction of neurons (mean ± s.e.m.) with significant choice selectivity ($p < 0.01$ with Bonferroni correction) in 9 non-overlapping 1-s epochs out of all choice-selective neurons. **d** Example PPC neurons that are choice selective during the ITI

accuracy on the remaining trials ('test set').

$$\log \frac{\text{probability}\{\text{choice}(N)=\text{forward}\}}{\text{probability}\{\text{choice}(N)=\text{downward}\}}$$
$$= w_s \cdot \text{stimulus}(N) + w_o \cdot \sum_{k=1}^{N-1} \text{outcome}(k) \cdot e^{-\frac{N-1-k}{\tau_o}}$$
$$+ w_c \cdot \sum_{k=1}^{N-1} \text{choice}(k) \cdot e^{-\frac{N-1-k}{\tau_c}} \quad\quad (1)$$
$$+ w_{oc} \cdot \sum_{k=1}^{N-1} \text{outcome}(k) \cdot \text{choice}(k) \cdot e^{-\frac{N-1-k}{\tau_{oc}}} + \text{constant}$$

This model with choice-outcome history predicted the behavior significantly better than the stimulus alone (Fig. 2a, b), indicating that the choice variability observed in this task is indeed not random. Instead, a significant part of the variability arises from a systematic influence of choice-outcome history that biases the decision on a trial-by-trial basis. To assess the contributions of distinct components of history information to the internal bias, we built partial models using only a subset of history information at a time (Fig. 2c). On average, the outcomes (reward or error) of previous trials carried the largest predictive power, followed by the choices (forward or downward) of previous trials. The weight of each variable and time constants in the full model (Eq. 1) were examined in sessions in which the corresponding variable exerted significant influence on the choice (Fig. 2d, e). We found that the weight for the visual stimulus was significant and positive in a majority of sessions (9/17), demonstrating that animals properly used the stimulus information despite a low behavioral performance. The weight for the previous trial outcomes was significant in 9/17 sessions with short (<1) time constants, indicating that mice tended to choose one

direction after reward and the other direction after error (e.g., forward following reward trials and downward following error trials, irrespective of the choices of previous trials). This strategy differs from the so-called 'win-stay/lose-switch' in which choice depends on both the outcomes and choices of previous trials (e.g., forward following rewarded forward trials, but downward following rewarded downward trials). Furthermore, the choices of previous trials significantly contributed to the prediction in 8/17 sessions, all with negative weights and longer (~10 trials) time constants, showing that mice had a tendency to equalize the frequencies of both forward and downward choices over time (Fig. 2d, e). In the non-memory version of the task in which mice performed better (Fig. 1c), the weight for visual stimulus was significantly larger, and the outcome history weight was significantly smaller than in the memory task (Supplementary Fig. 2b), indicating an enhanced use of the stimulus information and reduced importance of history information.

Mice employed somewhat common strategies in the memory task as described above, but the weights and time constants of individual terms were highly variable across individual sessions (Fig. 2d, e). Consistent with the variability of weights and time constants, model accuracy was significantly worse when a model built for one session was applied to another session of the same mouse (Supplementary Fig. 3). Thus, the rules that each mouse employed to use choice-outcome history were variable over days, similar to those previously reported in human subjects[5]. The detriment in model accuracy was even larger when a model from one mouse was applied to another mouse, demonstrating an idiosyncratic nature of the strategies (Supplementary Fig. 3). Taken together, the imperfect behavioral performance in conjunction with the behavioral model gives us an opportunity to estimate the hidden internal biases underlying decision variability on individual trials that are not directly measurable.

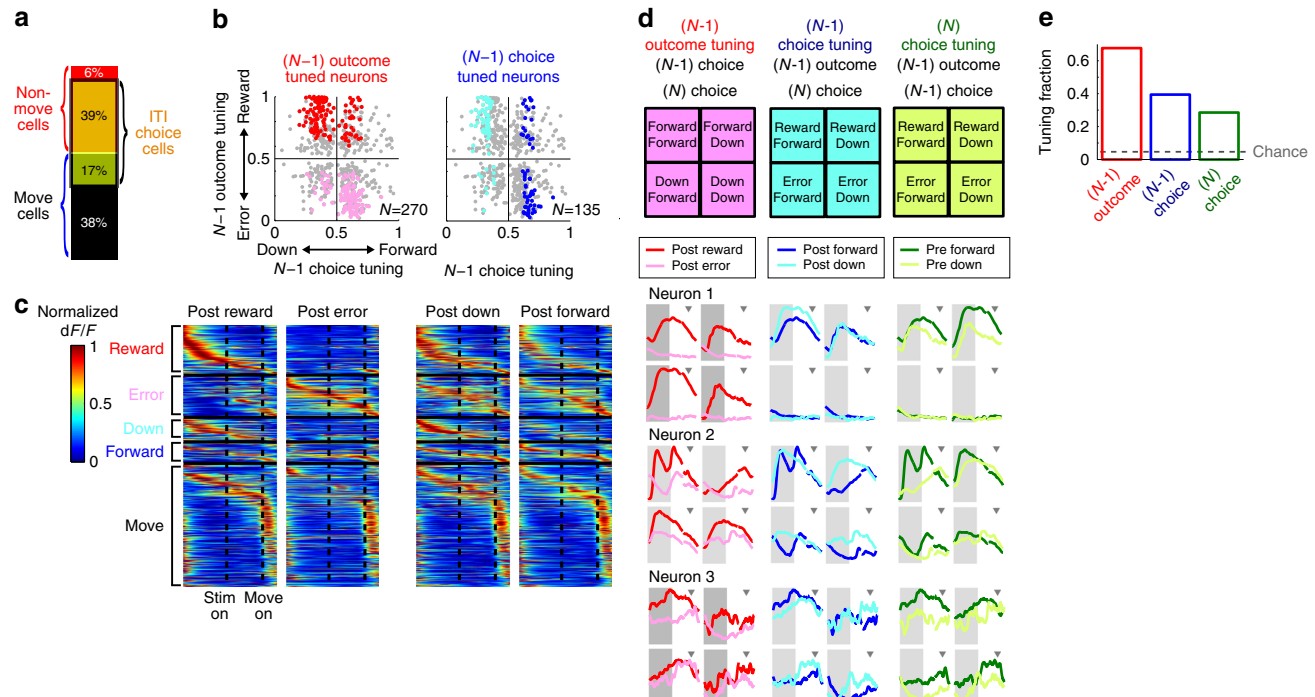

**Fig. 4** Pre-stimulus choice-selectivity in PPC encodes a mixture of history and bias information. **a** Break-down of choice-selective cells. The majority of cells that show significant choice-selectivity during ITI ('ITI choice cells') are different from those that show maximal choice-selectivity during peri-movement period ('Move cells', peri-movement period defined as −0.5 to 1.5 s around movement onset). **b** Neurons are tuned to the previous trial (N−1) choice and/ or outcome. Tuning was measured as the area under the ROC curve, ranging from 0 to 1. The value of 0.5 indicates no tuning. Dots are all choice-selective neurons (N = 652). Red dots are neurons with stronger activity during the ITI following reward (N = 146; 'Reward cells'), and pink dots are neurons with stronger activity following errors (N = 124; 'Error cells'). Blue dots are neurons with stronger activity following forward choice (N = 63; 'Forward cells'), cyan dots are neurons with stronger activity following downward choice (N = 72; 'Downward cells'), and gray dots are choice-selective neurons that do not show significant tuning to N−1 choice or outcome. **c** Trial-average activity of Reward, Error, Downward, Forward, and Move cells in their preferred choice direction trials under four different N−1 trial conditions. Each row represents the normalized activity of a single neuron sorted by its peak activity timing. **d** Previous trial (N−1) outcome, previous trial (N−1) choice, and upcoming trial (N) choice tuning of three example PPC neurons. Tuning for each variable is assessed in four different conditions in each of which the other two variables are fixed as indicated in the tables (top). In each condition, the mean traces of dF/F are plotted for two values of the variable of interest (red vs. pink, blue vs. cyan, or green vs. lime). PPC cells are influenced by all three variables in complex ways (e.g., cell 1 encodes N−1 outcome predominantly, yet N−1 choice and N choice also modulate its ITI activity). Gray shaded region indicates the ITI, and the triangle indicates the average movement onset time. **e** Fraction of cells whose ITI activity is significantly tuned for each of (N−1) outcome, (N−1) choice, or N choice, independent of the other two

We note that these strategies can be somewhat adaptive or maladaptive, due to our stimulus selection algorithm. Specifically, the same stimulus was repeated after error trials, and the stimulus was changed to the other stimulus after three consecutive rewarded trials with the same stimulus. In all other trials, stimulus was randomly selected. These rules were introduced to discourage mice from selecting the same choice (forward or downward) in every trial. Thus, the two common strategies described above, biasing choice based on the outcome history of the immediately preceding (N−1) trial, and equalizing choice frequencies, were both adaptive and helped mice perform better than chance. In contrast, a strong constant preference of one choice was an example of maladaptive strategies, as it results in repetition of the same error for multiple trials. The varying degrees of 'adaptiveness' of models in different sessions were quantified by assessing the success rate of the internal bias model (excluding the stimulus term in the full model) of each session in simulation in which the stimulus was selected according to the same rules. We found that 53% of sessions showed significantly adaptive strategies, while 24% were significantly maladaptive (Fig. 2f).

**The pre-trial activity of PPC represents internal biases.** To explore the neural basis of these subjective, history-based internal biases, we applied two-photon calcium imaging to record the neural ensemble activity in PPC while mice performed the task (Fig. 3a). We chose PPC because it is widely implicated in decision-making processes, PPC is highly interconnected with visual and motor areas, and PPC neurons encode a recent history of choice and outcome, placing it in an ideal location to bias the transformation of visual information to motor outputs[19, 21–23, 26, 27]. We imaged neurons in layer 2/3 expressing GCaMP6f in the 17 sessions from 8 mice whose behavioral analyses were presented in the previous section. 991 unique neurons (mean: 58/session, range: 15–123) showed significant task-related activity and were included in the analysis.

Consistent with previous recording studies in monkeys and rodents[21, 23, 28, 29], many PPC neurons (66%, 652/991) exhibited choice-selective activity such that the activity in forward-pressing and downward-pressing trials was significantly different. Choice-selective neurons presented varying timing of choice selectivity, and largely distinct populations of neurons showed choice selectivity during the stimulus, memory, and movement periods (Fig. 3b). Surprisingly, a large fraction of choice-selective neurons (56%, 362/652) showed significant choice selectivity during the inter-trial interval (ITI) before the stimulus was presented (Fig. 3b–d). In other words, these neurons had predictive

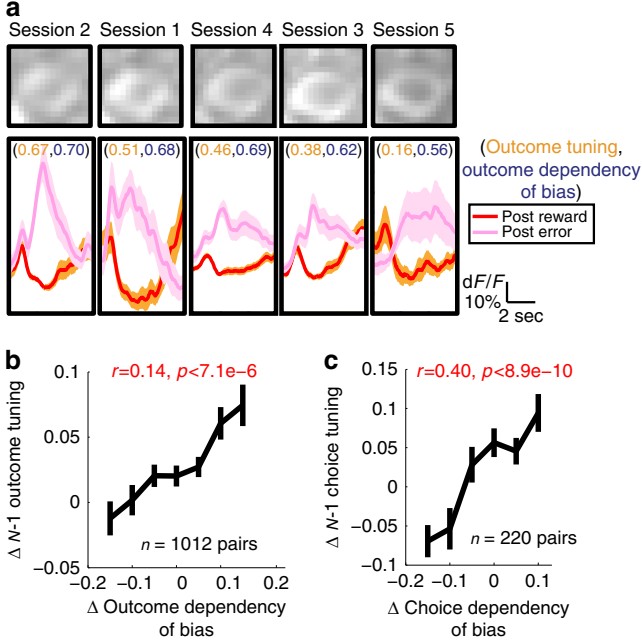

**Fig. 5** History tuning of PPC neurons reflects the history dependency of bias. **a** Top: a single neuron imaged in five separate sessions, sorted in the descending order of the (N−1) trial outcome tuning strength. Bottom: pre-stimulus activity (mean ± s.e.m.; −6 to 0 s from stimulus onset) of the example neuron following reward and error trials. Each pair of numbers represents the tuning strength of the neuron for (N−1) trial outcome and outcome dependency of the internal bias (estimated as the accuracy of the partial model using only outcome information), respectively. Note that the outcome tuning is stronger in sessions in which the past outcome information has stronger contributions to the bias. **b** Between-session differences in (N−1) outcome tuning of the ITI activity of single neurons plotted against between-session differences in choice dependency on outcome information (Pearson correlation coefficient = 0.14, $p < 7.1e-6$). Neurons with significant outcome tuning in at least one of the two sessions are included. **c** The same as **b** but for the (N−1) trial choice

information about the eventual choice of the mouse before the trial was initiated.

A trivial explanation for the ITI choice-selectivity is that the mice are already preparing or partially executing the movement during the ITI, and thus ITI choice-selective neurons are movement-related neurons. To address the possibility that mice are partially executing the movement during the ITI (e.g., leaning on the joystick), we performed additional experiments in which the joystick was unfixed in the ITI of a subset of trials. Choice-selectivity of PPC neurons during the ITI in the trials when mice did not apply force on the unfixed joystick remained the same as in the joystick-fixed condition, suggesting that the ITI choice-selectivity does not arise from partial execution of movements (Supplementary Fig. 4). Moreover, a majority of ITI choice-selective neurons (69%, 251/362) were non-movement neurons, distinct from movement neurons that showed maximal choice-selectivity during the peri-movement period (−0.5 to 1.5 s around movement onset; Fig. 4a). Conversely, a majority of non-movement choice-selective neurons showed choice-selectivity during the ITI (85%, 251/292). The choice-selectivity of these non-movement choice-selective neurons appears to be related to their sensitivity to the recent choice-outcome history. Of non-movement neurons, 92% (270/292) and 46% (135/292) were significantly tuned during the ITI to the outcome and choice of the immediately preceding (N−1) trial, respectively (Fig. 4b). Accordingly, these N−1 outcome- and choice-tuned neurons are

differentially modulated during ITI even when the choice in the upcoming N trial is the same, depending on the previous trial conditions (Fig. 4c).

Notably, many non-movement choice-selective neurons (43%, 125/292) were tuned to both N−1 outcome and N−1 choice, suggesting that multiple types of distinct history information are mixed at the level of individual neurons. However, in theory, the concurrent tuning to multiple variables could result from the correlation of N trial choice with both N−1 outcome and N−1 choice as shown by our behavioral modeling. To disentangle this confounding relationship, we evaluated how the ITI activity of individual ITI choice-selective neurons was modulated by history information (N−1 trial outcome and N−1 trial choice) and N choice (used as a binary estimate of internal bias) independently, by focusing on trials in which 2 of the 3 variables were identical. For example, neuron 1 shown in Fig. 4d was strongly modulated by N−1 outcome, exhibiting different levels of activity after rewarded and error trials. This modulation by N−1 outcome was clearly present even when we considered only the trials in which N−1 choice and N choice were fixed, indicating that the N−1 outcome modulation of this neuron was not a secondary effect of correlation between N−1 outcome and N−1 choice or N choice. In addition, this neuron was also modulated by both N−1 choice and N choice, independent from its modulation by N−1 outcome. Overall, large fractions of ITI choice-selective neurons exhibited independent tuning for N−1 outcome, N−1 choice, and N choice (Fig. 4e), indicating that distinct history and bias information is encoded in overlapping but distinct populations of individual PPC neurons.

Importantly, such tuning of individual neurons was not fixed but sensitive to the current strategies employed by the mice. We addressed this issue in a subset of experiments in which we imaged the activity of the same populations of PPC neurons across multiple sessions (386 neurons in 4 mice were imaged across 4–7 sessions). For example, the neuron in Fig. 5a imaged over 5 sessions is tuned to N−1 outcome in all imaged sessions during the ITI. However, the strength of its N−1 outcome tuning varied across sessions, tracking the strength of the influence of previous outcome on the subsequent choice as estimated by the accuracy of the partial model using the previous outcome information only. That is, the neuron showed more pronounced N−1 outcome tuning in sessions in which the previous outcome influenced the upcoming choice more strongly. Such flexible modulation of N−1 outcome tuning was consistent across PPC neurons (Fig. 5b). Similar effects were found for N−1 choice tuning (Fig. 5c). The flexible sensitivity of PPC neurons to distinct history information may underlie the flexible, subjective use of history to generate bias revealed by our behavioral modeling (Supplementary Fig. 3).

In line with this notion that PPC represents the subjective bias, the weighted sum of ITI activity of neurons simultaneously recorded from PPC was able to fit very closely the internal biases estimated by our behavioral model (Fig. 6a–c). The excellent fit was specific to the internal biases, and ITI activity could not fit the trial-shuffled internal biases (Fig. 6b). The PPC ITI activity closely tracked the fluctuations of the strengths of internal biases even in trials of the same choice (Fig. 6b, c), suggesting that the PPC ITI activity reflected continuously varying internal biases rather than categorical choice. PPC ITI activity better predicted the subsequent choice when the bias direction estimated from PPC ITI activity matched that of the subsequent stimulus, supporting the idea that the final choice is made by integrating the biases encoded in PPC and the subsequent stimulus (Fig. 6d).

The earlier analyses of single neuron responses showed that individual PPC neuron encode history and choice information in a mixed and heterogeneous manner (Fig. 4). Consistently, the

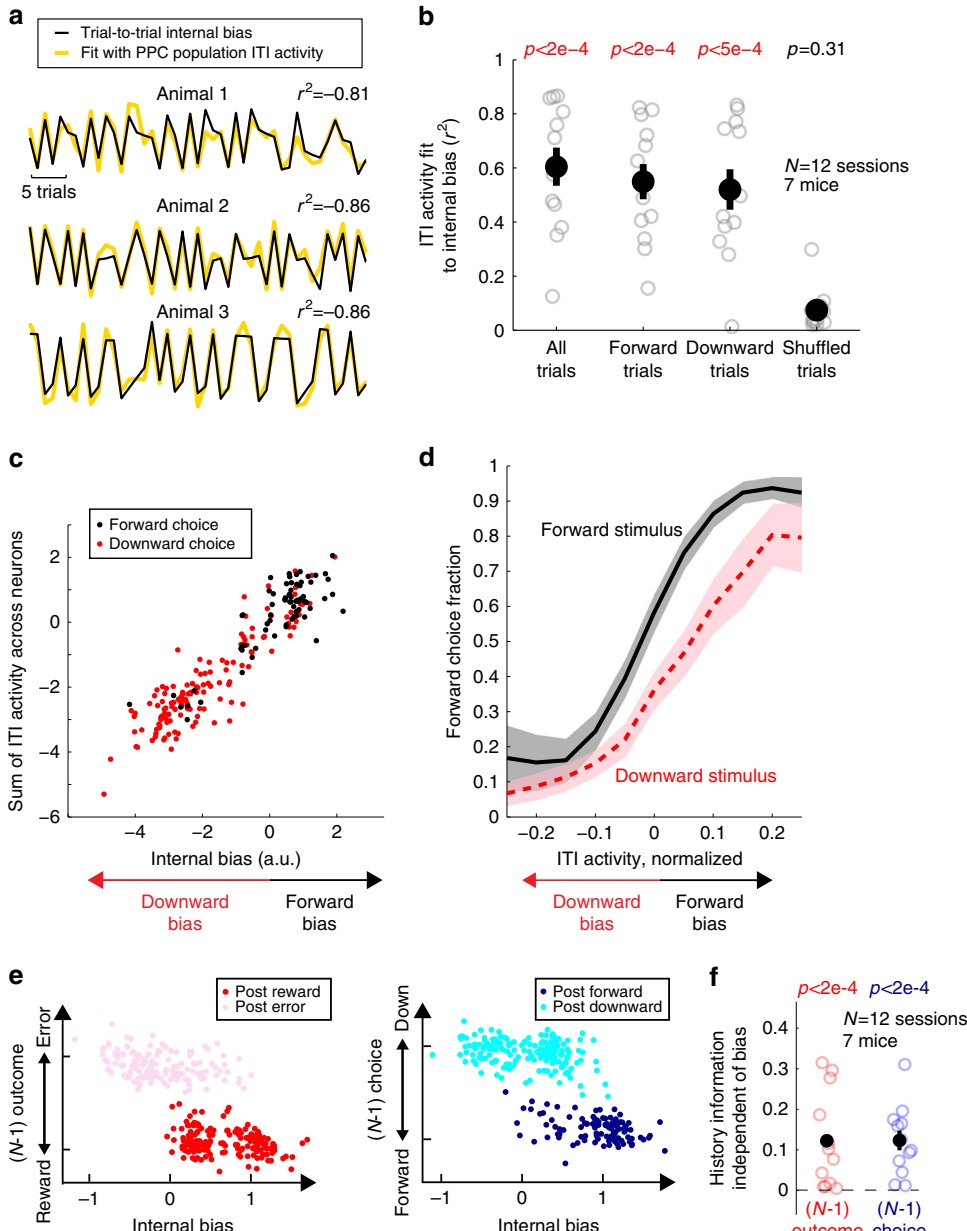

**Fig. 6** Pre-stimulus population activity in PPC reflects the history-dependent bias and previous trial history. **a** Sequences of internal bias (thin black lines, estimated from the behavioral model) and weighted sum of ITI activity (thick yellow lines) across simultaneously imaged PPC neurons in three example mice. **b** Goodness of fit of PPC ITI activity to the internal bias across (from left to right): all trials, only forward choice trials, only downward trials, and shuffled trials. Only the sessions in which the history-dependent bias is significant (likelihood-ratio test between the full model and a partial model that contains only stimulus and constant terms, $p < 0.05$, 12 sessions; Experimental Procedures) are included. Black, mean ± s.e.m. across sessions; gray, individual sessions. Wilcoxon one-sided signed rank test. **c** Data from a single session showing the weighted sum of PPC ITI activity across simultaneously imaged neurons against the internal bias estimated from the behavioral model. Each dot represents a single trial. **d** Fraction of forward choice as a function of the internal bias estimated from the ITI activity (mean ± s.e.m. across 12 sessions). Black, forward stimulus trials; red, downward stimulus trials. The estimated internal bias in each session was normalized such that it represents the signed distance from the decision boundary (Methods section) that divides the forward and downward choice trials. **e** Example population ITI activity from one session projected on a two-dimensional plane. Left: x-axis, activity encoding internal bias (i.e., estimated internal bias from the population activity, as in **a**), y-axis, activity encoding $N-1$ trial outcome (i.e., estimated $N-1$ trial outcome from the population activity), each dot, a single trial. Population activity in trials with similar internal bias still encodes $N-1$ trial outcome information (reward or error). Right: x-axis same as the left and y-axis shows the activity encoding $N-1$ trial choice. **f** History information independent of internal bias in the ITI activity, computed as (prediction accuracy of history information decoded from the ITI activity − prediction accuracy of history information decoded from the ITI activity projected to the internal bias axis). Each dot represents a single session in which the behavioral model predicts the choice sequence significantly ($p < 0.001$). Black, mean ± s.e.m. across mice. Wilcoxon one-sided signed rank test

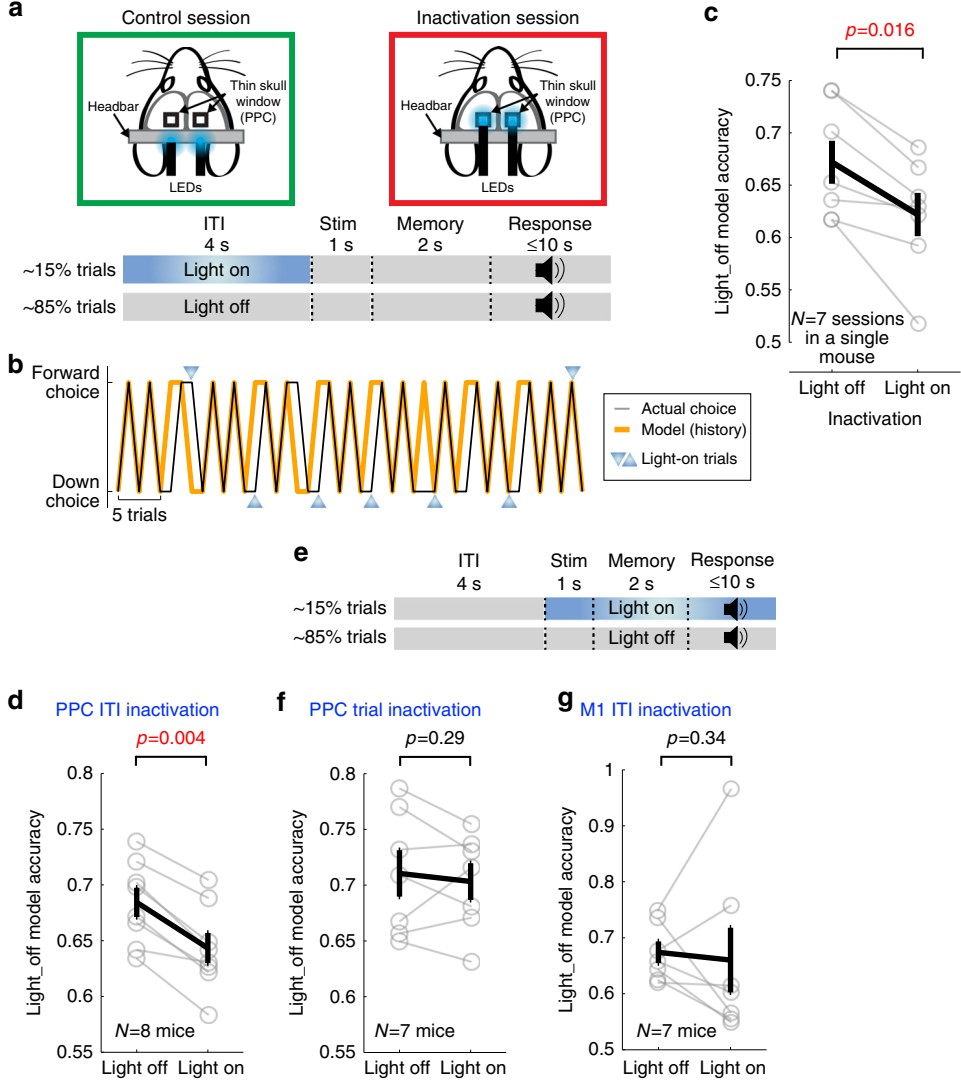

**Fig. 7** Inactivating pre-stimulus activity in PPC alters internal bias. **a** Schematic of inactivation experiment. Control (blue light directed away from PPC) and inactivation (the light directed to PPC) sessions alternated day-to-day (for 14–16 days). Continuous blue light was applied during the ITI in randomly selected trials (15%; light-on trials) in both control and inactivation sessions. **b** Choice sequence (black) and behavioral model fit (orange) in an example inactivation session. In this example, the mouse tended to alternate choice (i.e., the mouse most heavily weighted the previous choice history) in light-off trials, but this tendency was reduced in light-on trials. **c** The effect of PPC ITI inactivation on the model fit in seven separate inactivation sessions in a single mouse. Black, mean ± s.e.m. across sessions; gray, individual sessions. Wilcoxon signed rank test. The light-off model was built on a subset of light-off trials and its accuracy was assessed on the remaining light-off or light-on trials. **d** Average light-off model accuracy in light-off and light-on trials in inactivation sessions. Black, mean ± s.e.m. across mice; gray, individual mice. Wilcoxon one-sided signed rank test. **e** In trial inactivation sessions, blue light was applied from stimulus onset to the end of randomly selected trials (15%). Trial inactivation does not have a temporal overlap with ITI inactivation. **f** The same as **d**, but for trial inactivation in PPC. **g** The same as **d**, but for ITI inactivation in M1

PPC ensemble activity also encodes a mixture of history and bias information. When the ensemble ITI activity was fit separately to the internal biases and the outcome in the $N-1$ trial, we found that the population activity encoded both bias and $N-1$ outcome independently. That is, even for the same value of internal bias, population activity was still separable depending on the $N-1$ outcome (Fig. 6e, f). Similar results were found between bias and $N-1$ choice (Fig. 6e, f). Thus, PPC neuronal population encoded both history and bias information independently during the ITI.

**Inactivation of PPC ITI activity alters internal bias**. The results so far indicate that PPC contains information about action selection biases during the pre-stimulus ITI. To address whether

this information in PPC is indeed used to bias the subsequent actions, as opposed to the alternative possibility that actions are biased by activity elsewhere and PPC activity simply correlates with it, we used optogenetics to inactivate PPC during the task. We injected Cre-dependent AAV encoding Channelrhodopsin-2 (ChR2) in PV-Cre mice to express ChR2 in parvalbumin-positive inhibitory neurons in PPC and these mice were trained with the task ($N = 8$). Once their performance reached a plateau, we started inactivation sessions in which blue light was applied bilaterally to inactivate PPC during the ITI of a small subset (~15%) of trials (Fig. 7a). If PPC is indeed essential for the internal biases based on choice-outcome history, then PPC inactivation should alter the history dependency of choices. We tested this idea by building a behavioral model with a subset of the unperturbed light-off trials as the training set and testing the

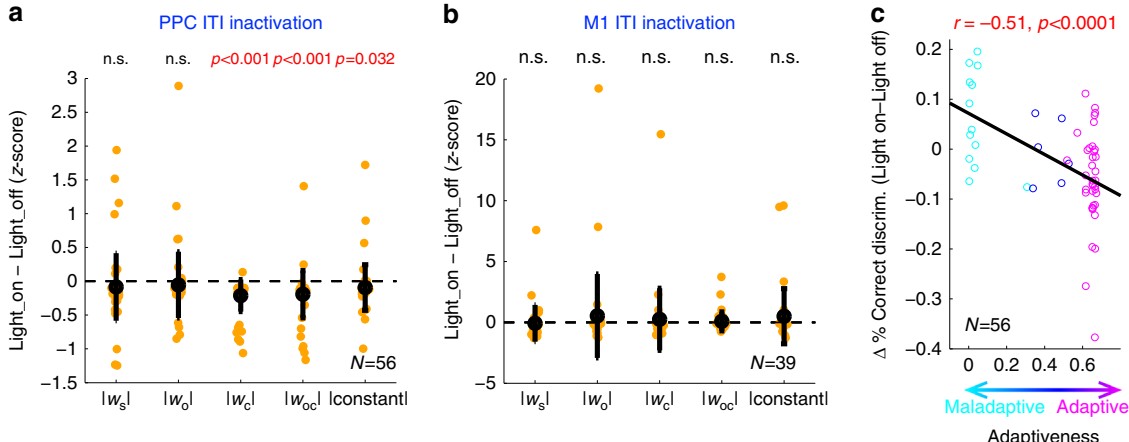

**Fig. 8** Inactivating pre-stimulus activity in PPC weakens the history dependency of choice. **a** Changes in the magnitude of model weights by PPC ITI inactivation. The weight magnitude of light-on model was translated into a z-score relative to the distribution of weight magnitudes of light-off models in each session. The weight magnitude for choice history, outcome-choice interaction history, and constant significantly decreased by PPC ITI inactivation (bootstrap, $p < 0.001$, $p < 0.001$, and $p < 0.05$ respectively). Black, mean ± s.d. across sessions. **b** The same as **a**, but for M1 ITI inactivation. **c** Change in behavioral performance during ITI inactivation (light-on–light-off) as a function of the degree of adaptiveness of the strategies (cyan: maladaptive, blue: neutral, and magenta: adaptive strategy sessions, respectively). The negative correlation between the performance change and the degree of adaptiveness indicates that ITI perturbation improves performance in sessions with maladaptive strategies, whereas it deteriorates performance in sessions with adaptive strategies. Black, linear regression

accuracy of choice prediction for the remaining unperturbed light-off and inactivated light-on trials. Consistent with our hypothesis, the model built with unperturbed trials ('light-off model') was significantly better at predicting the choice on other light-off trials than light-on trials (Fig. 7b–d and Supplementary Fig. 7). This result shows that PPC inactivation altered the idiosyncratic relationship between choice-outcome history and the subsequent actions. Such an effect was not observed in control sessions of the same mice in which the light was directed at the head bar instead of PPC (Supplementary Fig. 8a). Consequently, the light-off model performed significantly worse in inactivation light-on trials than control light-on trials, indicating that the effect was due to PPC inactivation and not due to non-specific effects of light. The altered history dependency occurred without significant changes in reaction time or movement time (Supplementary Fig. 7).

To examine the temporal specificity of inactivation effects, we inactivated PPC during the trial period, which starts from the visual stimulus onset and lasts until the end of a trial, thus not overlapping with the ITI (Fig. 7e; $N = 7$, subset of the PPC ITI inactivation mice). To our surprise, the history-choice relationship was not altered when we inactivated PPC during the trial period (Fig. 7f and Supplementary Fig. 8b). Accordingly, the model performed significantly worse in ITI inactivation trials than trial inactivation trials. These results suggest that the bias information encoded in PPC during the pre-stimulus ITI is subsequently maintained elsewhere to guide behavior independent of later PPC activity. Importantly, the consistent effect of ITI inactivation on internal bias was specific to PPC. When we inactivated the primary motor cortex (M1) during the ITI in a separate set of mice ($N = 7$), the effect was variable across animals (Fig. 7g and Supplementary Fig. 8c), thus no significant difference was observed between unperturbed light-off and inactivated light-on trials. Therefore, the altered relationship between history and subsequence choice is not a general effect of inactivation, but instead it is specific to PPC.

To delineate the nature of the altered relationship between history and choice by PPC ITI inactivation, we fit the light-on and light-off trials with separate full models (Eq. 1), and compared the

weights of the two models. Because of a greater number of light-off trials, we built light-off models using randomly sampled light-off trials matching the number of light-on trials 100 times. The light-on model was compared to the distributions of the 100 light-off models of each session. We found significant decreases in the weights for the choice history, outcome-choice interaction history, and constant (Fig. 8a). It is notable that the weight for outcome history did not change significantly although neural activity in PPC encodes the outcome history strongly (Fig. 4e). Therefore, outcome information may be redundantly represented in many areas and thus PPC inactivation alone does not alter the outcome dependence of choice, while PPC may be important more uniquely for previous choice information. In contrast, weights did not change in control sessions (Supplementary Fig. 8d, e), and the decreases in the three weights were significantly larger than in control sessions. ITI inactivation of M1 did not lead to significant changes in weights, either (Fig. 8b). These results indicate that PPC inactivation during the ITI weakened the dependency of subsequent action choice on choice-outcome history.

Given that some sessions showed adaptive and maladaptive strategies (Fig. 2f), we hypothesized that weakened history dependency would deteriorate behavioral performance in sessions with adaptive strategies, and improve performance in sessions with maladaptive strategies. Consistent with this prediction, the change in behavioral performance induced by PPC ITI inactivation was negatively correlated with the degree of adaptiveness of the strategies (Fig. 8c). In contrast, the change in behavioral performance and the degree of adaptiveness was not significantly correlated in control sessions (Supplementary Fig. 8f). Accordingly, the slope of linear regression on performance change against adaptiveness was significantly steeper in the inactivation sessions compared to control sessions (bootstrap; 56 inactivation vs. 106 control sessions; $p < 0.005$). These changes in behavioral performance provide additional evidence that PPC ITI activity is essential for the history-dependent biases, and the bidirectional effects suggest that PPC is responsible for the range of variable and idiosyncratic strategies to utilize the history information.

## Discussion

The pre-stimulus activity of PPC during the ITI closely reflected the subjective, internal bias estimated by our behavioral model and accordingly predicted the future choice. Choice-predicting pre-stimulus activity has been reported in various brain areas including the visual, parietal, premotor, and prefrontal cortex[21, 26, 30–33]. However, in contrast to the current study, these previous studies did not systematically relate the pre-stimulus activity to decision variables such as history-dependent internal bias and could not distinguish it from stochastic neural noise such as ongoing fluctuations of baseline. Moreover, the causal relationship between pre-stimulus activity and future choice has not been tested. To our knowledge, our current study is the first to demonstrate that the PPC pre-stimulus activity is essential for the influence of biases on subsequent actions.

Our temporally precise optogenetic inactivation revealed that the effect of PPC inactivation was specific to ITI, and perturbation after stimulus onset did not cause a measurable effect on behavior. This result implies that bias information encoded in PPC during ITI is unloaded to some other areas after the ITI and maintained in a PPC-independent manner. PPC neurons have projections to various brain areas[23, 34], and identifying these downstream areas that are responsible for the bias execution is an important topic of future research. We also note that our result does not imply that the post-stimulus choice-selective activity in PPC has no functions. In fact, several studies reported altered behavioral performance in perceptual discrimination tasks following PPC perturbation[27, 29, 35]. PPC also contributes to movement planning and execution, and inactivation in monkeys can affect movement end point control[36]. Such a role in fine motor control or sensory evidence accumulation is distinct from the bias coding that we describe here and was not tested in this study.

We found that the PPC ITI activity contains both history and bias information mixed at the level of individual neurons. This observation clearly excludes two extreme possibilities; (1) PPC only contains history information and is upstream of bias computation, and (2) PPC only contains bias information and is downstream of bias computation. While the precise circuit mechanisms underlying the transformation of history information into bias are extremely difficult to uncover, based on the mixed representation of history and bias in PPC, we favor the view that PPC participates in the computation of subjective bias from history information. It is important to note that these PPC neurons that encode history and bias information are intermingled with other neurons that are selectively active during visual stimulation, delay, and movement periods. PPC thus likely contains multiplexed, parallel pathways dedicated to the processing of distinct forms of information.

The functional homology between primate and rodent PPC is not fully established, but several rodent PPC studies have found neural response properties analogous to primate PPC and started to provide further insights into PPC circuits and functions[23, 27, 35, 37, 38]. Especially, two recent findings in rodent PPC resonate with our current study: (1) PPC population activity exhibits slow dynamics that integrate recent events[22], and (2) PPC perturbation affects internally guided decisions[39]. Our finding that PPC neurons encode a mixture of history and bias to influence action selection demonstrates an important functional consequence of the former observation. The representation of history-dependent internal bias in PPC presents a mechanism for PPC to affect internally guided decisions. Furthermore, our finding that intermingled but distinct sets of neurons represent specific sets of information lays foundation for investigating functional diversity in PPC microcircuits, likely linked with projection target areas.

## Methods

**Animals**. All procedures were in accordance with protocols approved by the UCSD Institutional Animal Care and Use Committee and guidelines of the National Institute of Health. Mice (calcium imaging: cross between *Gad2-IRES-Cre* [JAX 010802][40] and *Rosa26-CAG-LSL-tdTomato* [JAX 007914][41] or *Rosa26-CAG-LSL-tdTomato* or cross between Camk2a-tTA [JAX 003010] and tetO-GCaMP6s [JAX 024742]; optogenetic perturbation: *PV-Cre* [JAX 008069][42] or cross between *PV-Cre* and Ai32 [JAX024109]) were housed in a room with a reversed light cycle (12–12 h). Experiments were performed during the dark period.

The animal sample size was determined based on previously published studies, and no randomization or blinding were applied when allocating animals to experimental groups.

**Long-term behavioral training**. Adult mice (six weeks or older, male and female) were implanted with a custom head-fixation plate on the skull. Following a minimum 3 days of recovery, daily water consumption was limited to a controlled amount (typically 1 mL/day). Behavioral training began following 3–10 days of water restriction.

A custom-built behavioral apparatus housed in a box (40 × 40 × 40 cm) included a joystick (M11L061P; CHProducts), a 17 inch computer monitor (for visual stimulus presentation; placed ~15 cm from the right eye of the mouse), and a water port with photodiodes to sense licking (Fig. 1a). The stock joystick handle was custom machined and retrofitted with a 1/16 inch thick brass rod that mice manipulated with their left forepaw (Supplementary Fig. 1a). An electromagnet (EM050–3–222; APW) was situated so it could be used to mechanically immobilize the joystick at the origin. The joystick had a dynamic range of 56° in each angular direction forming a spherical endpoint space (Supplementary Fig. 1a). The 2D angular position of the joystick was continuously recorded at 1 kHz using a data acquisition card (USB6008; National Instruments) and custom Matlab software. The task-sequence execution, stimulus selection, auditory cue presentation, reward dispensation, and task time recording were coordinated by an open source real-time Linux/Matlab software package BControl (http://brodywiki.princeton.edu/bcontrol/). The presentation of visual stimuli (100% contrast, full-field, square wave drifting gratings 0.04 cycles/degree, and 3 cycles/sec) was implemented using Psychtoolbox (an open source Matlab toolbox; http://psychtoolbox.org/).

In the two-alternative forced-choice task (Fig. 1a), one of two orthogonal visual stimuli (forward or downward moving gratings) was presented for 1 s, followed by a 2-sec memory period. After the memory period, an auditory cue (6 kHz pure tone) marked the response period (up to 10 s) during which the joystick entering the correct target area (hereafter referred to simply as 'target'; Supplementary Fig. 1a) in the same direction as the gratings triggered a water reward. Errors (i.e., entering the incorrect target, and movements before the go cue) triggered a white noise sound and led to an immediate trial-abortion. Following reward, trial-abortion, or no response, the return of the joystick to the origin ended the trial and initiated an ITI (4 or 8 s, constant within each session). During the ITI the joystick was immobilized at the origin by an electromagnet. At the end of the ITI (simultaneous with the beginning of visual stimulus onset), the electromagnet was disengaged, and the joystick was free to move. Thus, if mice already pushed the joystick in any direction from the ITI, the joystick would have moved out of the origin as soon as the next trial stimulus period began and the trial was most likely aborted (see below for the withholding requirement). However, these movements immediately after the ITI (movement onset within 100 ms from stimulus onset) were rare (1%, 45/4747).

Mice were trained under head-fixation in the behavioral apparatus, ~1 h per day over a period of 2–4 months. The task was shaped to reach the final version through 8 training steps (Supplementary Fig. 1b). In the first step, the mice received a water reward as long as they moved the joystick to the correct target within a 30-s response period (regardless of whether or not they hit the incorrect target first). As they became more proficient with pressing the joystick in both directions, we increased the target distance from 6.7° (~6 mm) to 11.1° (~10 mm). In step 2, we decreased the response time to 10 s, and trained the mice until they reached the targets during the 10 s response period in more than 80% of trials. In both steps 1 and 2, the joystick was mechanically fixed by the electromagnet until the auditory go cue.

In step 3, to prevent mice from pushing or leaning on the joystick before the go cue, we released the joystick from electromagnet immobilization simultaneously with visual stimulus onset, and rewarded the mice only if they moved the joystick during the response period (i.e., withheld movements until after the go cue) and reached the correct target. Trials in which mice responded before the go cue were considered errors and immediately aborted, resulting in a white noise error sound. Step 3 training continued until mice achieved withholding performance above 80%:

$$\text{Witholding performance} = (\text{Number of responding trials after go cue})/$$
$$(\text{Number of all responding trials})$$

In step 4, mice were trained to discriminate between the two distinct visual stimuli (forward and downward drifting gratings) and reach the correct target after the go cue. In this step, trials were considered errors and immediately aborted if mice reached the incorrect target, or moved before the go cue (as in step 3). step 4

continued until they achieved both withholding and discrimination performance above 80%:

$$\text{Discrimination performance} = (\text{Number of trials hitting the correct target})/$$
$$(\text{Number of trials hitting any target})$$

Discrimination performance was computed for all trials that reached a target regardless of whether or not the trials were successfully withheld. Once this performance criterion was achieved, the ITI length was gradually increased to 4 or 8 s (step 5). In step 6, we turned off the visual stimulus during the response period (i.e., visual stimulus was turned off simultaneously with the go cue). In step 7, the stimulus period was shortened to 1.8 s and a 0.2 s memory period was introduced. In the final step, the visual stimulus period was gradually decreased to 1 s and the memory period was gradually increased to 2 s. With the 2-s memory period, the discrimination performance rarely improved above 60% (even after prolonged training). Thus, we trained each mouse until their discrimination performance in the 2-s memory task reached 60% on average.

A subset of mice performed sessions containing randomly interleaved non-memory and memory trials (both with a 3-s pre-movement period between the stimulus onset and the go cue; in non-memory trials, the visual stimulus stayed on until a target was reached) during training. In those sessions, the discrimination accuracy was consistently lower in memory than non-memory trials, indicating that the memory load, rather than the sensory discriminability, impaired performance in the memory task (Fig. 1c).

**Visual stimulus**. The visual stimulus was randomly selected between forward or downward drifting gratings with the following constraints: (1) after three consecutive rewarded trials in one direction, the stimulus always switched to the other direction, and (2) after error trials, the same stimulus was repeated. These constraints were implemented to discourage the mice from choosing only one direction and settling at 50% discrimination accuracy. Despite the deterministic stimulus after an error or a third consecutive reward in one direction, the mice performed only slightly better in those trials than random trials (Supplementary Fig. 1c), indicating that mice did not fully utilize these hidden stimulus presentation rules to their advantage.

Because of the pseudo-random rules of stimulus presentation, the fraction of correctly discriminating trials (# of trials hitting the correct target/# of trials hitting any target) achieved by random choice would not be 50% if there was a constant choice preference. So we estimated the constant choice preference within a session and converted it to a probability to choose each choice using the following formulae:

$$\text{Probability of choice 1} = \frac{1}{2} \times \left( \frac{\text{Number of trials}_{\text{choice 1 | stimulus 1}}}{\text{Number of trials}_{\text{stimulus 1}}} + \frac{\text{Number of trials}_{\text{choice 1 | stimulus 2}}}{\text{Number of trials}_{\text{stimulus 2}}} \right)$$

$$\text{Probability of choice 2} = 1 - \text{Probability of choice 1}$$

Then, the chance level performance for the given session was computed by simulating random binary choice with the estimated probabilities under the same pseudo-random rules 1000 times (Fig. 1b).

**Behavioral model**. In our behavioral model, the choice on a given trial is predicted by a weighted sum of the current stimulus, the history of past trial outcome, choice, and their interaction, and a constant (Eqs. 1 and 2). Past trials were temporally discounted in an exponentially decaying manner (i.e., stronger effect from more recent trials) with time constants fit independently for each history variable. Stimulus, outcome, and choice were all binary variables with the value of 1 or −1. However, in trials in which mice did not reach a target, choice was zero and outcome was 1 (error).

We repeated the following procedure for a fixed set of time constants (varying from 0.01 to 100 for each history variable), and selected the time constants and weights that produced the highest model accuracy as the best-fit regression parameters. For given time constants, we found best weights using logistic regression on a training set (Eq. 1), and then estimated the choice sequence in a designated test set using the best weights (Eq. 2). The two-step process was 10-fold cross-validated. That is, trials within a session were divided in 10 non-overlapping parts, where each part served as a test set once, and the other nine parts as a training set. The fit of the model (or simply, model accuracy) was measured as the fraction of test trials in which the estimated choice matched the actual choice.

$$\widehat{\text{choice}}(N) = \left. \begin{cases} 1, & \text{if } p > 0.5 \\ -1, & \text{otherwise} \end{cases} \right\} \qquad (2)$$

In partial models, we used a subset of variables and performed the same regression procedure. For instance, when estimating the effect of inactivation on trial-history dependency of choice, we used a partial model without the stimulus term and compared the partial model accuracy between light-on and light-off trials.

To assess the statistical significance of history information in predicting future choices, we applied a likelihood-ratio test between the full model and a partial model that contains only stimulus and constant terms. We used $p < 0.05$ as a significance threshold.

To determine whether the specific history-dependent strategy of a given session was adaptive or not (Fig. 2f), we generated a sequence of choices following the estimated history model (i.e., the partial model without the stimulus term) and the same stimulus rules described earlier. After simulating 100 sequences, if the fraction of correctly discriminating trials was >0.5 in more than 95% of the iterations, the strategy was classified as adaptive. If the fraction of correctly discriminating trials was <0.5 in more than 95% of the iterations, it was maladaptive. In the other cases, the strategy was neutral.

**Imaging neural activity**. After mice reached the discrimination threshold of 60% in the 2-s memory task, we paused training and allowed water access at least for 2 days prior to craniotomy and virus injections. The craniotomy spanned both the PPC (stereotaxic coordinates relative to bregma: 1.7 mm lateral, 2.0 mm posterior) and the forelimb region of the primary motor cortex (M1; stereotaxic coordinates relative to bregma: 1.5 mm lateral, 0.3 mm anterior) in the right hemisphere. Viruses (AAV2-1-hSyn-GCaMP6f diluted in saline 1:7, or AAV2-9-hSyn-GCaMP6f diluted in saline 1:7; UPenn Vector Core) were injected at 5 sites (~20 nL per site) in PPC and M1 at a depth of ~250 μm beneath the dura, in layer 2–3. After the injections, the craniotomy (~2 mm × 3.5 mm) was covered with an optical window fixed in place with dental cement. Two of the three mice in the free-joystick task condition (Supplementary Fig. 4) were generated by crossing Camk2-tTA and tetO-CGaMP6s and received the same procedures for craniotomy and optical window implant without virus injections.

Following surgery and recovery (14–35 days after the surgery), we imaged cortical activity in layer 2–3 at the depth of ~200 μm with excitation at 925 nm from a Ti–Sa laser (Spectra-physics) using a two-photon microscope (B-scope, Thorlabs). Each imaging field was 512 × 512 pixels covering 472 × 508 μm and imaging was performed at ~28.4 Hz. The duration of each behavior-imaging session limited to 1.5 h, ended when the mouse was disengaged from the task, or completed 170 rewarded trials. Mice completed ~135 (range: 88–172) rewarded trials in each imaging session.

For each mouse, 1–4 different imaging fields were studied within PPC (one field per session). For some mice ($N = 4$), the same fields were imaged repeatedly over 4–7 sessions. Of the repeatedly imaged fields, except for the analysis tracking selectivity for immediately preceding trial outcome and choice across sessions (Fig. 6), we included only the session with the most number of active cells.

**PPC inactivation experiment**. Mice for PPC inactivation experiments (PV-Cre; $N = 8$) were implanted with a head-fixation bar and bilaterally injected with virus carrying ChR2 (AAV2-1-EF1A-DIO-hChR2, undiluted; UPenn Vector Core) through a thinned skull over PPC. Approximately 100 nL of virus was injected in one location at each of two depths, 200 μm and 600 μm from the dura. After the surgery, following the same training protocol as the head-plate implanted mice described above, we trained them to perform the task over a period of 2–4 months.

Once a mouse reached the 60% discrimination criterion, we inspected the previously thinned-skull area and performed re-thinning if necessary. Then, we conducted 1–7 light acclimation sessions to minimize non-specific light effects on behaviors. In the acclimation sessions, bifurcated blue LED fibers (470 nm, 11–20 mW for inactivation in each fiber, Doric) were placed ~2 mm above the head-fixation bar, away from the cortical region expressing Channelrhodopsin, and lights were turned on during the ITI of randomly selected 15% of trials. Most mice recovered their previous task performance within 1–2 days.

Each inactivation experiment was performed across 14–16 daily sessions. Control and inactivation sessions alternated day-by-day for all but 5 mice. The 5 mice (3 trial inactivation and 2 ITI inactivation) performed control and perturbation sessions sequentially in 7-day blocks. In control sessions, the LED lights were directed above the head-fixation bar, whereas in inactivation sessions they were placed directly above PPC on both hemispheres. Except for this difference, all procedures were identical between control and inactivation sessions.

In both control and inactivation sessions, light-on trials were pseudo-randomly selected with a constraint that there be at least 5 light-off trials between any two adjacent light-on trials to avoid potential behavioral adaptation to cortical perturbation due to consecutive and/or frequent exposures. Under this restriction, light stimulation was applied to ~15% of trials.

**M1 inactivation experiment**. The procedures were identical to the PPC inactivation experiment described above except for the following difference. In three of the seven mice, ChR2 was expressed in PV positive neurons by crossing PV-Cre mice with Ai32 mice containing lox-stop-lox-ChR2 in the ROSA locus, and an optical window was placed over M1. We did not observe behavioral differences during the inactivation experiment between these three mice and the rest.

**Trial selection**. In behavioral model analyses, choice was predicted only for trials in which mice reached any of the two targets after the stimulus onset, ~248 trials (range: 137–344; 91%) per session. In neural data analyses, we included trials in which mice reached any target within 1 s after the go cue. Error trials in which mice moved the joystick before the go cue (~15% of trials) were excluded due to the possibility that neural activity during stimulus and memory period in those trials might be contaminated with immediate movement planning and execution. Slow trials (target acquisition taking longer than 1 s from the go cue; ~16%) were also excluded to reduce neural variability associated with highly dissimilar movement kinematics within the same categorical choice. By these criteria, ~185 trials per session (range: 92–293) were analyzed. The early and late trials excluded from the neural analysis showed similar choice tuning to the regular trials (Supplementary Fig. 5).

**Single-cell activity**. Using custom Matlab program, fluorescence images were aligned frame by frame to compensate for lateral motions. Regions of interest (ROIs) were manually drawn on the motion-corrected fluorescence images, by circumscribing the cell bodies based on their GCaMP fluorescence intensity distinguishable from the background. Pixels inside each ROI were considered as a single cell, whereas pixels extending radially outward from the cell boundary by 2–6 pixels were considered background. In case the background included other cells' ROIs, those pixels were excluded. To estimate the activity of a single cell, 70% of the average pixel intensity in its background was subtracted from the average pixel intensity inside the cell[43]. The time series of the background-adjusted intensity was transformed to dF/F by dynamically estimating the baseline intensity (i.e., the 8th percentile of the intensity distribution in the 20 s window centered at each time point)[23]. For GCaMP6s signals recorded to compare ITI tuning between free-joystick and fixed-joystick conditions, dF/F was further transformed into an estimate of spike rates using the spike-triggered mixture model (https://github.com/lucastheis/c2s)[44].

**Active cells**. To detect calcium transients, we used a zero-mean dF/F trace in which the mean dF/F was subtracted from the original dF/F. Using Matlab function findpeaks, we first identified tentative transient peaks. If the amplitude of a detected peak was at least 0.5 and >3.3 times the standard deviation of dF/F velocity per frame, the peak was counted as a calcium transient. To focus our analysis on stable and reliable cells, we only included the cells that showed calcium transients at a rate >1 transient/minute in both the first and second half of a session and the average peak amplitude of all transients is greater than five times the standard deviation of dF/F. By these criteria, the average number of analyzed cells (or, active cells) in a single PPC field was 73 (range: 22–140).

**Task-related cells**. Of the active cells, we identified task-related cells that showed significant activity modulation during the task as following. The mean activity trace of each neuron was calculated by aligning dF/F traces to behavioral events and averaging across all correct trials. Three different behavioral events were used to align dF/F traces: stimulus onset (−6 to 3 s), movement onset (−2 to 7 s), and reward onset (−4 to 5 s). A cell was considered to be task-related if its mean activity fell outside the 99.9th percentile of its dF/F distribution in three consecutive frames in any of the three alignments. For this criterion, the false positive rate estimated on temporally-shifted dF/F traces, by a random amount for each trial, was 4.4%.

**Choice-selectivity in trial epochs**. For the task-related cells, their choice-selectivity was examined in 9 non-overlapping 1-s epochs (Fig. 3): the first 6 epochs aligned to stimulus onset (−4 to 2 s), and the latter 3 epochs aligned to movement onset (−1.5 to 1.5 s). The ITI started ~1.3 s after movement onset on average. To obtain both selectivity strength and significance, we performed receiver operating characteristic (ROC) analysis on the time-averaged activity in each epoch, using the binary choice as label and the activity as score. For a given area under the ROC curve (AUROCC), double the distance from 0.5 (i.e., $2 \times |\text{AUROCC}-0.5|$) was taken as the selectivity strength. For a significance test, we used the 99.9th percentile of the null distribution of selectivity strength ($p < 0.01$ with Bonferroni correction for multiple comparisons) estimated by choice label shuffling per cell and epoch, 1000 times. The preferred directions of choice-selective neurons were nearly equally distributed (Supplementary Fig. 6).

**Selectivity for other behavioral variables**. Selectivity or tuning for other behavioral variables (e.g., $N−1$ trial outcome and choice) was computed in the same way as choice selectivity, but with those binary variables as the label or score (Figs. 4 and 5).

**Fitting trial-by-trial internal bias with PPC ITI activity**. We used a linear regression to fit the trial-by-trial fluctuation of internal bias with the trial-by-trial ITI population activity, following 10-fold cross-validation method (Fig. 6a). To further avoid overfitting, cells that significantly contribute to the linear regression of the internal bias were selected using Matlab function *stepwisefit* on all trials, before applying linear regression. The fit was measured as $r^2$ achieved on test sets.

**Linear classifier**. To compute the prediction power of the neural activity on binary behavioral variables such as ($N−1$) trial outcome, ($N−1$) choice, and N choice, we computed the classifying accuracy of a linear classifier (Fig. 6f). The weights and constant of each classifier were estimated by a logistic regression represented in Eq. 3 on a training set, and its performance was evaluated on a test set using Eq. 4, following the standard 10-fold cross-validation method. The classifier performance was defined as the fraction of test trials in which the prediction matched the actual variable. ITI population neural activity is high dimensional (dimension = N cells × 4 ITI epochs/cell = 4N). Thus to avoid overfitting, only the features (i.e., selective epochs of selective cells) that significantly contributed to the regression were selected using Matlab function *stepwisefit* using all trials before applying the classification analysis.

$$\log \frac{\text{probability}\{\text{behavior} = 1\}}{1 - \text{probability}\{\text{behavior} = 1\}} = w \cdot \text{predictor}(N) + \text{const} \quad (3)$$

$$\widehat{\text{behavior}}(N) = \begin{array}{ll} 1, & \text{if} \quad p > 0.5 \\ -1, & \text{otherwise} \end{array} \quad (4)$$

**Neural distance from the decision boundary**. We computed the signed Euclidian distance of the ITI population activity from the linear decision boundary of the N trial choice classifier for each trial (Fig. 6d). That is, the distance for the activity on one side of the boundary was positive, and the other side negative. Given the strong correlation between the PPC ITI activity and the internal bias estimated from our behavior model, the neural distance serves as a proxy for the strength of internal bias.

**Decoding $N−1$ trial history information independent of internal bias**. To estimate the amount of history information independent of internal bias in the population ITI activity, we built two classifiers. The first classifier decoded $N−1$ outcome information from the population ITI activity. The second classifier decoded $N−1$ outcome from the internal bias related activity (i.e., the weighted sum of the population ITI activity that best fit the internal bias estimated from our model). Then, to compute the outcome information independent of the internal bias, we subtracted the accuracy of the second classifier from the first (Fig. 6f). The independent $N−1$ choice information was computed similarly.

**Choice selectivity independent of immediately preceding trial outcome and choice information**. Because the immediately preceding ($N−1$) trial outcome and choice information had predictable power for the upcoming trial choice, we inspected whether the ITI activity remains choice selective even for the trials that followed the same outcome and choice conditions in the $N−1$ trial. Thus, the ITI choice selectivity was examined in each of the four possible conditions of the $N−1$ trial: (1) post-reward and post-downward, (2) post-reward and post-forward, (3) post-error and post-downward, and (4) post-error and post-forward (Fig. 4d). The ITI activity would not be N choice selective in any of these 4 conditions if its activity purely encoded $N−1$ outcome or choice because the $N−1$ outcome and choice were the same within each condition. To ensure statistical power, we examined the choice selectivity of cells only for the conditions with at least 18 trials per choice direction (118 neurons with one condition and 65 neurons with two conditions). In each of these conditions, choice selectivity of each cell was assessed using ROC analysis. The fraction of conditions in which the cells are choice selective was compared to the null distribution estimated by shuffling choice labels in each condition.

**History information tuning independent of N trial choice**. Similarly to the choice selectivity independent of history information described above, tuning for $N−1$ trial outcome was assessed in four different conditions, in which $N−1$ choice and N choice were fixed. Likewise, tuning for $N−1$ choice was assessed in four conditions in which $N−1$ outcome and N choice were fixed.

**Excluded sessions in the analysis of inactivation effects on the behavioral model weights**. In some behavioral sessions, animals chose only one direction in the light-on trials. Because a numerical solution of logistic regression on a sequence of constant choice cannot produce a reliable estimate for weights, we removed these sessions from the weight change analysis (8 sessions across 3 mice of M1 ITI inactivation and 3 sessions across 3 mice of control).

**Movement analysis**. Movement onset was defined as the first time at which the joystick velocity exceeded 22.2°/sec (~20 mm/s) continuously for 20 msec and the joystick moved at least 1.3° (~1.1 mm) from the origin. The reaction time was measured as the time from the go cue and movement onset, and the movement duration was measured as the time from movement onset to the time when the joystick entered any target region (Supplementary Fig. 7).

**Statistical analysis**. Throughout the paper, we performed non-parametric tests to avoid normality assumption. Similarity in variance between groups was not explicitly tested. When simply assessing whether the medians of paired samples acquired from the same subjects are different, we used Wilcoxon signed rank test. When testing for unpaired samples acquired from different subjects, we used Wilcoxon rank sum test. When testing a specific hypothesis that the median of one set of samples is greater (or smaller) than the median of the other set acquired from the same subjects, we used Wilcoxon one-sided signed rank test. For two sets of samples acquired from different subjects, we used Wilcoxon one-sided rank-sum test for testing specific hypotheses. For statistical tests for means, we used bootstrapping.

**Bootstrapping for the statistical test of model parameter changes**. To examine whether the mean change in a parameter across $N$ different inactivation sessions is significantly different from zero, we randomly drew $N$ values from the original $N$ observations allowing repetitions, computed the mean of $N$ random samples, and constructed the probability distribution of the mean through 1000 repetitions. When the 95% confidence interval of the distribution did not include zero, we deemed the mean significantly different from zero.

**Data availability**. The data that support the findings of this study are available from the corresponding authors upon reasonable request.

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

## Acknowledgements

We thank A. Mitani for custom image acquisition settings and motion correction software; A.N. Kim, K. O'Neil, L. Hall, and T. Loveland for technical assistance; B. Danskin, L. Maggioni, A. Hoang, T. Vu, K. Zhang, H. Chen, D. Riccardo, S. Sadre, S. Lu, C. Wong, T. Zubatiy, R. Makin, Y. Htay, L. Phan, Q. Fujii, and E Lu for behavioral training, the GENIE Project at Janelia Research Campus for GCaMP6f, D. Lee, J. Leutgeb, R. Malinow, C. Niell, M. Wilke and members of the Komiyama lab for comments and discussions. This work was supported by grants from NIH (R01 NS091010A, R01 EY025349, R01 DC014690, P30 EY022589, and U01 NS094342), Japan Science and Technology Agency (PRESTO), Pew Charitable Trusts, David & Lucile Packard Foundation, McKnight Foundation, Human Frontier Science Program and New York Stem Cell Foundation to T.K.; J.E.D. was supported by the NRSA fellowship (F31NS090858). T.K. is a NYSCF-Robertson Investigator.

## Author Contributions

E.J.H. and T.K. conceived the project. J.E.D. developed the behavioral apparatus and task. M.M. performed the optogenetic experiments. All other experiments were performed by

E.J.H. and analyzed by E.J.H. and T.K.; E.J.H. and T.K. wrote the manuscript with inputs from J.E.D. and M.M.

## Additional information

**Competing interests:** The authors declare no competing financial interests.

