## [Peer Review File · Nature Communications]

Editorial Note: This manuscript has been previously reviewed at another journal that is not operating a transparent peer review scheme. This document only contains reviewer comments and rebuttal letters for versions considered at Nature Communications. Editorial Note: Parts of this peer review file have been redacted as indicated.

Reviewers' comments:

Reviewer #1 (Remarks to the Author):

The paper has improved considerably over the course of numerous rounds of review, especially in the way it casts the story, and in the way it compares the effect of M1 inactivation to PPC inactivation. Overall, it tells elements of an interesting story. But it tells the story in a complicated way (especially in the figures), and the interpretation of the inactivation data is unclear. Surely the data will be of interest to people who study parietal cortex and related cortical regions, but it is not clear that the paper can reach a wider public.

The main criticisms are that (1) the analysis of PPC responses (figures 3, 4, 5, 6) is laborious and intricate (only the most dedicated readers will make it past figure 4), and (2) the analysis of inactivations is unsatisfactory. Indeed, though the inactivation measures are potentially interesting, they are mostly cast in terms of the effect of inactivation on model accuracy, which is not easy to interpret. When model accuracy goes down, we simply don't know what the mouse is doing. The only graphs about inactivation that seem interpretable are those in Figure 8b (effects of PPC inactivation on model parameters) and supplementary figure 8e (effects of M1 inactivation on model parameters).

This reviewer is frankly not available for further advice on this paper (having reviewed it multiple times) but in future versions, the main suggestion is to go back to the main text (which is more readable than the figures) and look at the points it makes, and keep only the figures/panels that are needed to support those points. And find a way to tell us what inactivation does to the mouse, besides making a model behave worse. If that's shown in Figure 8b, and indeed all sessions are in that graph (no data selection), show us Figure 8b, show us Supplementary Figure 8e, and that's all we need to see. Flooding the paper with countless graphs make the paper lose clarity and impact.

A more general suggestion for future work: For a broader and clearer picture one would have wanted a model of PPC responses that takes into consideration the various predictors (previous outcome, previous choice, etc). A GLM model would probably be the first thing to try. But that's for a different paper.

SPECIFICS

Figure 2. panel d is not described in legend. The next panel over is not given a name (and is described as d in the legend). Panel e is missing an x label.

Figure 3c. It would be appropriate to show the same figure for the other choice (the non-preferred one). To avoid circularity, the assignment of the preferred choice to each neuron should be done with a different data set from the one used in these plots.

Figure 3. Legends for panels d and e seem to have been swapped. Also, not clear why we need to see 4 examples in panel b and then 12 examples in panel e. Can't we just see the 12 examples in panel e?

Figure 4d is utterly mysterious, starting with the matrices at top. How are the colors related to the colors in b and in the legends under d? Only the most motivated readers will want to understand what is happening here, and the conclusion seems to be vague: "PPC cells are influenced by all three variables in complex ways".

Figure 5. A detail: in panel a there is no need to use color to distinguish the sessions. They are

distinguished by their position in the page. There is already enough color in all the plots.

Figure 6. b. Why do we need all these conditions (all trials, forward, trials, etc). It's a lot to pay attention to. Do we need panel b at all?

Figure 6 c: the ordinate shows a weighted sum. Were the weights chosen from these very same data? If so, no wonder the correlation is so high. The analysis is circular.

Figure 6d-f. Panel d is a bit mysterious. Panels e and f: this reviewer has lost the willpower to understand what they show. Other readers may be in the same situation. What's the point of these analyses?

Figure 7d-g. These figures show the effect of inactivation on model accuracy, not the effect of inactivation on behavior. So when model accuracy goes down, we simply don't know what the animal is doing. This is not informative. We need to know how inactivation affects model parameters, or the relative quality of two different models.

Same for figure 8a. It is about model accuracy. The only place where we see the effect of inactivation on model parameters (on what the mouse is doing) is in Figure 8b. It would be nice to show that there are no similar effects with inactivation of area M1 (which is presumably shown in Supplementary Figure 8e, though the legend does not say).

Reviewer #2 (Remarks to the Author):

I have been a reviewer on this paper previously and will re-state my enthusiasm for the novelty of some of the findings in this paper, especially that the animal's choice bias is smaller when there is disruption of PPC during the ITI. I also appreciated how the authors are now leveraging the animal's imperfect performance as a way to understand how choice bias can affect decision-making. Finally, the authors' conclusions that PPC's neuronal population encodes history and bias information independently during the ITI is important and fits with other recent observations about task-relevant variables being represented independently in PPC even at the level of single neurons. Taken together, these findings mean that the paper will be of interest to many readers.

Major comments:

1. Figure 2d: Why are only 9 points included when there were 17 sessions? The legend for this figure only talks about the right panel (“(d) Time constant of each history variable.”, line 463). Does the explanation of which points are included apply to the points selected for inclusion in the left panel, too? I can understand why certain points are excluded for the right panel: if there was no effect of choice history, of course it doesn't make sense to include the time constant. But for the left panel, we really need to see all the points and have significance indicated by shading.

2. The fact that the changes following optogenetic disruption were limited to w_c and w_{oc} (and not w_o) is pretty important. This constrains the kinds of computations that are being done in PPC since the behavior was affected by both choice and outcome. Also, what should we make of the fact that w_{oc} did change but w_o did not? The authors need to flesh this out. I appreciate their honesty about the mistake in the previous version, but this new finding does change things.

Minor comments:

1. Line 65: It would be more natural to describe the grating as moving "rightward or downward" as opposed to "forward and downward." Forward sounds like it is moving towards the mouse.
2. I was confused by Supp. Fig. 1c; I eventually tracked down its explanation in the methods in the main text, but it might be helpful to have additional information in the supplementary figure legend. My initial interpretation was that this was that "random stimulus" (label on abscissa) referred to a random relationship between stimulus and reward, as opposed to a truly random stimulus sequence.
3. The authors should really state actual p-values, especially for values that are close to the threshold like $p < 0.04$ and $p < 0.05$ (line 260 and Supp. Fig. 1c)

Reviewer #1 (Remarks to the Author):

The paper has improved considerably over the course of numerous rounds of review, especially in the way it casts the story, and in the way it compares the effect of M1 inactivation to PPC inactivation. Overall, it tells elements of an interesting story. But it tells the story in a complicated way (especially in the figures), and the interpretation of the inactivation data is unclear. Surely the data will be of interest to people who study parietal cortex and related cortical regions, but it is not clear that the paper can reach a wider public.

The main criticisms are that (1) the analysis of PPC responses (figures 3, 4, 5, 6) is laborious and intricate (only the most dedicated readers will make it past figure 4),

The paper has become extensive during the long revision process. In particular, the analysis of neural recording data has become intricate while addressing many concerns by the reviewers. As a matter of fact, figures 4-6 mainly address the original concern raised by this Reviewer about whether the PPC activity simply reflects ongoing movement or movement plan. Since other readers may have similar questions, it is prudent to keep these figures.

and (2) the analysis of inactivations is unsatisfactory. Indeed, though the inactivation measures are potentially interesting, they are mostly cast in terms of the effect of inactivation on model accuracy, which is not easy to interpret. When model accuracy goes down, we simply don't know what the mouse is doing. The only graphs about inactivation that seem interpretable are those in Figure 8b (effects of PPC inactivation on model parameters) and supplementary figure 8e (effects of M1 inactivation on model parameters).

We respectfully disagree. As we stated in the previous rebuttal, the decrease in model accuracy demonstrates that PPC inactivation alters the relationship between choice-outcome history and choice bias. This is the main result of the inactivation experiments that establishes a causal link between PPC activity and choice bias. (We suspect that perhaps this reviewer is mistaken about what Figure 7 shows. Perhaps, the reviewer thinks that we are comparing the accuracy of models built with light-on trials vs. the accuracy of models built on light-off trials, in which case the results are not very informative. Instead, we are building a behavioral model using a subset of light-off trials and testing the accuracy of the model on the remaining light-off trials vs. light-on trials. In this case, the lower accuracy on light-off trials indicates that PPC inactivation altered the behavior compared to light-on trials.) After we establish this change in Figure 7, we proceed to understand how the relationship changes in Figure 8.

This reviewer is frankly not available for further advice on this paper (having reviewed it multiple times) but in future versions, the main suggestion is to go back to the main text (which is more readable than the figures) and look at the points it makes, and keep only the figures/panels that are needed to support those points. And find a way to tell us what inactivation does to the mouse, besides making a model behave worse. If that's shown in Figure 8b, and indeed all sessions are in that graph (no data selection),

show us Figure 8b, show us Supplementary Figure 8e, and that's all we need to see. Flooding the paper with countless graphs make the paper lose clarity and impact.

Again, many panels were added during the nearly two years of revision process to address the reviewers' concerns. They are important in case other readers have similar questions.

A more general suggestion for future work: For a broader and clearer picture one would have wanted a model of PPC responses that takes into consideration the various predictors (previous outcome, previous choice, etc). A GLM model would probably be the first thing to try. But that's for a different paper.

This is obviously beyond the scope of this manuscript.

SPECIFICS

Figure 2. panel d is not described in legend. The next panel over is not given a name (and is described as d in the legend). Panel e is missing an x label.

We have corrected these points.

Figure 3c. It would be appropriate to show the same figure for the other choice (the non-preferred one). To avoid circularity, the assignment of the preferred choice to each neuron should be done with a different data set from the one used in these plots.

This panel is to show the time course of activity, so it does not need to include the non-preferred choice. However, we have included it in the revision for the symmetry. There is no point in defining choice-selective cells using only a subset of trials.

Figure 3. Legends for panels d and e seem to have been swapped. Also, not clear why we need to see 4 examples in panel b and then 12 examples in panel e. Can't we just see the 12 examples in panel e?

We have corrected the legends, and kept only the 12 examples.

Figure 4d is utterly mysterious, starting with the matrices at top. How are the colors related to the colors in b and in the legends under d? Only the most motivated readers will want to understand what is happening here, and the conclusion seems to be vague: "PPC cells are influenced by all three variables in complex ways".

The conclusion quoted by the reviewer precisely refutes the original concern by this reviewer that PPC cells may only reflect ongoing or planned movement. The colors in b and d are not related but clearly defined in each figure and its legends.

Figure 5. A detail: in panel a there is no need to use color to distinguish the sessions. They are distinguished by their position in the page. There is already enough color in all the plots.

We have removed the colors.

Figure 6. b. Why do we need all these conditions (all trials, forward, trials, etc). It's a lot to pay attention to. Do we need panel b at all?

Panel b is an important summary plot. Different conditions are important as they show that PPC activity is not categorical for two choices but is continuously modulated according to the internal bias. We now moved old Figure 6b to Supplementary Figure.

Figure 6 c: the ordinate shows a weighted sum. Were the weights chosen from these very same data? If so, no wonder the correlation is so high. The analysis is circular.

As has been clarified in our previous rebuttal and in Methods, all our regressions were done with 10-fold cross validation and not circular.

Figure 6d-f. Panel d is a bit mysterious. Panels e and f: this reviewer has lost the willpower to understand what they show. Other readers may be in the same situation. What's the point of these analyses?

Fig. 6d shows that 'the final choice is made by integrating the biases encoded in PPC and the subsequent stimulus' (quoted from main text). Fig. 6e-f show that 'PPC neuronal population encoded both history and bias information independently during the ITI' (quoted from main text), refuting the original concern of Reviewer 2 that PPC may only reflect a single scalar variable.

Figure 7d-g. These figures show the effect of inactivation on model accuracy, not the effect of inactivation on behavior. So when model accuracy goes down, we simply don't know what the animal is doing. This is not informative. We need to know how inactivation affects model parameters, or the relative quality of two different models.

We disagree. These results establish that PPC inactivation alters the relationship between choice-outcome history and choice. This is the main result that draws a causal link between history representation in PPC and choice bias.

Same for figure 8a. It is about model accuracy. The only place where we see the effect of inactivation on model parameters (on what the mouse is doing) is in Figure 8b. It would be nice to show that there are no similar effects with inactivation of area M1 (which is presumably shown in Supplementary Figure 8e, though the legend does not say).

Supplementary Figure 8e is control sessions (blue light away from brain) as stated in the legend. Supplementary Figure 9 is M1 sessions as stated in the legend. We moved the M1 inactivation result (previously Supplementary Fig. 9) next to PPC inactivation result in Fig. 8, and removed

the original Fig. 8a.

Reviewer #2 (Remarks to the Author):

I have been a reviewer on this paper previously and will re-state my enthusiasm for the novelty of some of the findings in this paper, especially that the animal's choice bias is smaller when there is disruption of PPC during the ITI. I also appreciated how the authors are now leveraging the animal's imperfect performance as a way to understand how choice bias can affect decision-making. Finally, the authors' conclusions that PPC's neuronal population encodes history and bias information independently during the ITI is important and fits with other recent observations about task-relevant variables being represented independently in PPC even at the level of single neurons. Taken together, these findings mean that the paper will be of interest to many readers.

Major comments:

1. Figure 2d: Why are only 9 points included when there were 17 sessions? The legend for this figure only talks about the right panel (“(d) Time constant of each history variable.”, line 463). Does the explanation of which points are included apply to the points selected for inclusion in the left panel, too? I can understand why certain points are excluded for the right panel: if there was no effect of choice history, of course it doesn't make sense to include the time constant. But for the left panel, we really need to see all the points and have significance indicated by shading.

We apologize for the oversight and added the legend for the left panel. The previous version only showed the data points in which the weights were statistically significant. We now include the non-significant points in grey.

2. The fact that the changes following optogenetic disruption were limited to w_c and w_{oc} (and not w_o) is pretty important. This constrains the kinds of computations that are being done in PPC since the behavior was affected by both choice and outcome. Also, what should we make of the fact that w_{oc} did change but w_o did not? The authors need to flesh this out. I appreciate their honesty about the mistake in the previous version, but this new finding does change things.

We agree that this is interesting. We speculate that outcome information may be redundantly represented in many areas and thus PPC inactivation alone does not alter the outcome dependence of choice. PPC may be important more uniquely for previous choice information. We have included these speculations in the revised manuscript. However, as our statistical power is limited due to the small number of inactivation trials, we would like to limit our interpretation of this negative result (lack of effect on w_o).

Minor comments:

1. Line 65: It would be more natural to describe the grating as moving “rightward or downward” as opposed to “forward and downward.” Forward sounds like it is moving towards the mouse.

From the perspective of the mice, these directions are forward (the direction mouse would walk towards) and downward (towards their feet). We hope this is sufficiently clear with the schematics in Figure 1.

2. I was confused by Supp. Fig. 1c; I eventually tracked down its explanation in the methods in the main text, but it might be helpful to have additional information in the supplementary figure legend. My initial interpretation was that this was that “random stimulus” (label on abscissa) referred to a random relationship between stimulus and reward, as opposed to a truly random stimulus sequence.

We apologize for the confusion and expanded the legend for Fig. S1c.

3. The authors should really state actual p-values, especially for values that are close to the threshold like $p < 0.04$ and $p < 0.05$ (line 260 and Supp. Fig. 1c)

We have included the actual p-values.

REVIEWERS' COMMENTS:

Reviewer #2 (Remarks to the Author):

I am satisfied with the reviewers' responses to my comments and I recommend this paper for publication as it is.

I do not agree with the other reviewer's criticism that the paper has become long and convoluted. The authors have been diligent in responding to extensive and diverse revision suggestions and were required to include many additional analyses as a result of this. The idea that the authors' diligence constitutes "flooding the paper with countless graphs" is patently unfair. My view is that these additional analyses have strengthened the paper, and have put the authors in a stronger position to argue for a role for PPC in incorporating recent history into current decisions. Further, I found that presentation of the data to be overall very clear, and to have become more clear over the course of the review.

This paper has the potential to be of interest to many readers and should be published immediately. The idea of altering trial history effects via PPC disruption is something a lot of labs are thinking about, and this group has moved quickly and carefully to demonstrate a clear role for PPC. This paper is an absolute fit for Nature Communications and I give it my full support.

[Redacted]

REVIEWERS' COMMENTS:

Reviewer #2 (Remarks to the Author):

I am satisfied with the reviewers' responses to my comments and I recommend this paper for publication as it is.

I do not agree with the other reviewer's criticism that the paper has become long and convoluted. The authors have been diligent in responding to extensive and diverse revision suggestions and were required to include many additional analyses as a result of this. The idea that the authors' diligence constitutes "flooding the paper with countless graphs" is patently unfair. My view is that these additional analyses have strengthened the paper, and have put the authors in a stronger position to argue for a role for PPC in incorporating recent history into current decisions. Further, I found that presentation of the data to be overall very clear, and to have become more clear over the course of the review.

This paper has the potential to be of interest to many readers and should be published immediately. The idea of altering trial history effects via PPC disruption is something a lot of labs are thinking about, and this group has moved quickly and carefully to demonstrate a clear role for PPC. This paper is an absolute fit for Nature Communications and I give it my full support.

We thank the reviewer for the kind words and also for the work as a reviewer [Redacted].